# Computer-Aided (In Silico) Modeling of Cytochrome P450-Mediated Food–Drug Interactions (FDI)

**DOI:** 10.3390/ijms23158498

**Published:** 2022-07-31

**Authors:** Yelena Guttman, Zohar Kerem

**Affiliations:** Institute of Biochemistry, Food Science and Nutrition, The Robert H. Smith Faculty of Agriculture, Food and Environment, The Hebrew University of Jerusalem, Rehovot 76100, Israel; elena.guttman@mail.huji.ac.il

**Keywords:** CYP, food–drug interactions, dietary compounds, virtual screening, databases

## Abstract

Modifications of the activity of Cytochrome 450 (CYP) enzymes by compounds in food might impair medical treatments. These CYP-mediated food–drug interactions (FDI) play a major role in drug clearance in the intestine and liver. Inter-individual variation in both CYP expression and structure is an important determinant of FDI. Traditional targeted approaches have highlighted a limited number of dietary inhibitors and single-nucleotide variations (SNVs), each determining personal CYP activity and inhibition. These approaches are costly in time, money and labor. Here, we review computational tools and databases that are already available and are relevant to predicting CYP-mediated FDIs. Computer-aided approaches such as protein–ligand interaction modeling and the virtual screening of big data narrow down hundreds of thousands of items in databanks to a few putative targets, to which the research resources could be further directed. Structure-based methods are used to explore the structural nature of the interaction between compounds and CYP enzymes. However, while collections of chemical, biochemical and genetic data are available today and call for the implementation of big-data approaches, ligand-based machine-learning approaches for virtual screening are still scarcely used for FDI studies. This review of CYP-mediated FDIs promises to attract scientists and the general public.

## 1. Introduction

Cytochrome (CYP) 450 enzymes are heme mono-oxygenases responsible for various functions in the human body. The catalytic process of CYPs involves the insertion of one atom of oxygen into an organic substrate while another oxygen atom is reduced to water. In the liver and the digestive system, CYPs are responsible for the phase I metabolism of a wide range of endogenous compounds: steroid hormones, lipids and bile acids, as well as xenobiotics, including prescribed drugs, environmental pollutants and dietary phytochemicals [1,2]. To facilitate their removal, xenobiotics are predominantly biotransformed by a limited subset of P450 enzymes, namely CYP1A2, CYP2A6, CYP2B6, CYP2Cs (principally CYP2C9 and CYP2C19), CYP2D6, CYP2E1 and CYP3A4, with some of these enzymes playing a larger role than others [3,4].

CYP3A4 is recognized as the main enzyme in drug metabolism in the liver and, no less importantly, in the gut. It accounts for approximately 80% of the total CYP content of the intestine and over 30% of the total CYP content of the liver [5]. Although the total amount of CYP3A expressed in the human small intestine represents approximately 1% of the amount expressed in the liver [6], substantial drug extraction takes place during the absorption of orally administered drugs [7,8,9]. This is due to the relatively high enterocytic drug concentration and the considerably lower blood flow to the intestine in comparison to the liver, which allows for the prolonged exposure of swallowed drugs to intestinal metabolizing enzymes [10]. The predominance of CYP3A4 in the human intestine and the higher concentrations of orally administrated drugs enable CYP3A4 to act several-fold more efficiently in the intestine than in the liver [5]. CYP3A4 alone is involved in the metabolism of over 50% of all marketed drugs [11]. Therefore, the potential interactions between promising new drugs and major CYP isoforms are assessed repeatedly from the earliest stages of the development of every new drug [12,13].

Accumulated evidence indicates the potent inhibition of CYPs, mainly CYP3A4, by dietary phytochemicals, many of which are consumed as spices, supplements and medicinal herbs [11]. Due to its pivotal role in drug metabolism, CYP inhibition or activation might impair drug clearance and increase the risk of toxicity or ineffectiveness, respectively. Examples of metabolic food–drug interactions (FDI) and herb–drug interactions (HDI) include those of furanocoumarins in grapefruit (*Citrus × paradisi*), stilbene *t*-resveratrol in grape skins (*Vitis vinifera*), black pepper (*Piper nigrum*), *Ginkgo biloba* and other herbs and spices that contain acclaimed and established bio-active agents with anti-hypertensive drugs, chemotherapeutics, serotonin analogs and others [11,14,15,16,17,18]. While the inhibiting compounds in some foods were identified, for others, the specific active compound is still unknown.

In rodents, the isoform CYP3A4 is expressed predominantly in the liver, with only scant expression in the intestine [19,20]. The differences in isoforms and distinct expression levels between P450s in human and rodent intestines limit the relevance of rodents as models to predict drug metabolism and oral bioavailability in humans [21]. Hence, studying the effects of ingested dietary compounds on intestinal drug metabolism in humans requires models other than rodents, such as cell cultures, microsomes and transformed microorganisms that express a CYP of interest or an array of CYPs [22,23,24,25,26,27].

Data on the plant-derived modifiers of CYPs has been slowly accumulating for two decades. The required time and resources for in vitro and in vivo assays limit the rate of discovery of new CYP inhibitors and, more specifically, the rate of discovery of dietary phytochemicals that might impair drug metabolism. In recent years, chemo-informatic, machine-learning and, more recently, deep-learning approaches have been used to identify the relationships between the structure and chemical properties of small molecules and their biological activities. These methods allow the rapid and efficient virtual screening of large chemical databases to identify compounds with the activity of interest. Other structure-based methods, such as docking and molecular dynamics (MD), allow in-depth studies of receptor–ligand and enzyme–inhibitor interactions. These approaches have contributed to our understanding of the mechanisms that underlie the biological activity in question. 

A vast number of published in silico studies have addressed drug–drug interactions (DDI) by developing focused virtual tools and identifying the synthetic inhibitors of CYP. These methods, which are described in this review, can be applied for the prediction of CYP-mediated FDI and HDI, as well as DDI. Moreover, careful consideration of the applicability domains of various models may allow the use of established tools to virtually screen the databases of dietary, food-derived and herbal compounds for CYP inhibitors. To that end, the in silico predictions of protein–ligand interactions may be divided into two major categories—(1) structure-based and (2) ligand-based—as will be discussed, in detail, below.

The importance of FDI mediated by CYPs at two separate sites, the intestine and the liver, calls for evidence-based data that should be made available to both professionals and the public. Such data are essential to support decision-making, drug development and dietary recommendations. Large databases of drugs, compounds in food, and bioactive natural products, together with cloud analytics, which have become available and user-friendly, provide opportunities to screen for potential FDI. Here, we review computational tools and approaches that have been used to predict CYP-mediated FDI.

## 2. Structure-Based Methods

Human CYPs are large membrane-associated flexible proteins and, therefore, present challenges for structural studies. The first published crystal structures of a human CYP, CYP2C9 (1OG52, 1OG5), were published in 2003 [28]. The first structures of CYP3A4 (1W0G, 1WOF, 1WOE and 1TQN) were published in 2004 [29,30], followed by the structures of additional important members of the CYP family [31,32,33,34]. Earlier models relied on homology to determine the structures of *Bacillus megaterium*, *Saccharopolyspora erythraea* and, to some extent, the structures of enzymes from rodents [35,36]. Currently, 139 crystal structures of the main human CYPs that are involved in drug metabolism, both unbound and bound to substrates or inhibitors, are available in the Protein Data Bank. The majority of the published structures are of CYP3A4, demonstrating the tremendous interest in this isoform (Figure 1). Indeed the published structures have greatly contributed to our understanding of which compounds bind to CYPs and how they do so [37].

### 2.1. Modeling the Mechanisms of CYP Action and Enzyme Inhibition

Crystal structures show that the active sites of CYPs are usually large, flexible and buried inside the protein. Therefore, ligands, both substrates and inhibitors, must enter and exit the active site via ligand tunnels, triggering conformational changes in the protein structure [38,39]. MD, which simulates the dynamic behavior of both the protein and the ligands in an aquatic environment, is often used to reveal the mechanisms underlying CYP activity and CYP inhibition. Kingsley and Lill (2015) [40] developed an MD-based method to demonstrate the ligand-induced tunnel flexibility of CYP2B6. Fischer and Smieško (2019) [41] used MD to show that substrates can access the active site of CYP2D6 through the protein–membrane interface. Mustafa et al. (2019) [42] identified structural differences that contribute to the differences in the substrate specificities of CYP2C9 and CYP2C19 by affecting the protein–membrane orientation, which affects substrate access tunnels. 

In classical biochemistry, the term ligand is used to describe a signal molecule that binds to a receptor and the term substrate refers to a molecule that is processed by an enzyme. In computational structural biology, a ligand is any small molecule bound to a protein; therefore, both substrates and inhibitors are referred to as ligands, although CYP is an enzyme rather than a receptor. The modeling of the change that the ligand undergoes during enzymatic activity poses a great challenge and is usually ignored for the sake of simplicity, unless the mechanism of enzymatic action is within the scope of the research project [43,44].

The active site of CYP3A4 contains spatially distinct substrate-binding domains within the enzyme’s active site [45], which allows for unique binding and kinetic behavior toward some ligands. The binding of a first ligand increases the affinity of CYP3A4 for the binding of a second ligand. Such a mechanism was observed in vitro in the binding of ketoconazole, a well-known inhibitor of CYP3A4 [46]. 2V0M is a crystal structure that demonstrates the simultaneous binding of ketoconazole in the catalytic pocket of CYP3A4 [47]. Later, an MD simulation combined with free-energy calculations was performed to elucidate the physicochemical origin of this positive homotropic cooperativity of ketoconazole–CYP3A4 binding [48]. 

CYPs, especially CYP2D6 and CYP2C9, are highly polymorphic enzymes. Genetic polymorphisms in CYP may affect enzyme structure and subsequent functioning and, therefore, may be responsible for inter-individual and inter-ethnic variation in FDI. 

Bioinformatic methods based on sequence homology, amino acid properties and phylogenetic trees have been developed to predict the impact of a mutation on protein structure and/or function [49]. However, web-based tools such as SIFT, PyloPhen and MutPred have performed poorly in the prediction of the functional relevance of CYP2B6. The use of a combination of tools and the molecular docking of substrates was shown to improve prediction accuracy, although some variants may not be correctly predicted by any publicly available tool [50].

Structure-based methods have been successfully used and combined to predict the effects of mutations on protein structure, stability, function and ligand binding [51,52]. Docking scores and poses have revealed variability in ligand interactions with wild-type and mutant proteins [53,54]. The combination of docking and MD simulations was used to demonstrate the differential binding of phytocannabinoids to the variant CYP2D6*17, as compared to WT CYP2D6 [55]. Another MD comparative analysis of four variants of CYP2D6 was performed to investigate the possible structural basis of CYP2D6 inhibition by a known agonist. Subsequent tunnel analysis of substrate access and solvent channels revealed varied bottleneck radii [56]. Dynamic in silico simulations can highlight cases in which the entire structure of the protein changes, sometimes even due to a single residue mutation, as well as changes in structural flexibility that may influence the activity of the mutated enzyme [56,57,58]. 

Alanine scanning is another computational method that can be useful for identifying mutations in CYP that may have clinical outcomes. This method determines the importance of amino acids in internal interactions between residues and protein–ligand interactions by recalculating the binding score before and after the replacement of each residue with alanine [59,60].

### 2.2. Prediction of Candidate Inhibitors

CYP inhibition can be either irreversible (also termed mechanism-based inhibition or suicidal inhibition) or reversible [13,61]. Irreversible inhibition requires the metabolism of the inhibitor by the enzyme, whereas reversible inhibition can take place directly, without metabolism [62,63]. Inhibitors can bind directly or in close proximity to the heme group or at distant allosteric binding sites in the protein [64,65]. These features make it rather challenging to design models to predict the inhibition of CYP based on its structure. However, attempts have been made and have proven successful. 

The common guiding principle of structure-based predictions is the use of a combination of independent computing methods to produce a single model. The first approach of choice is usually docking, which is quick, easy to use and to interpret, and also cheap in terms of computational power. However, docking lacks the ability to achieve the precision desired for a prediction model. The combination of docking with other structure-based methods, such as pharmacophore modeling and MD simulations, or ligand-based methods, in some cases, takes into account options such as multiple binding modes, protein flexibility and the large volume of active sites. Rossato et al. (2010) [66] used a combination of MD simulation, pharmacophore pre-alignment and docking protocol to establish an approach to predict ligand binding to CYP2C9 and CYP2D6. Using a pharmacophore model followed by docking, Zhu et al. (2011) [67] virtually screened ~1000 herbal compounds to find five inhibitors of CYP1A2. Joshi et al. (2017) [68] performed virtual screening for CYP1A1 inhibitors using a combination of a pharmacophore model and sequential docking steps with different levels of precision.

The resolved structures or docking poses of inhibitors at the binding site can be used to model a three-dimensional quantitative structure–activity relationship (3D-QSAR), using comparative molecular-field analysis (CoMFA) or comparative molecular-similarity indices analysis (CoMSIA) methods. The basic idea of CoMFA/CoMSIA is that the shapes of the non-covalent fields surrounding the molecules are often related to their biological properties and are, therefore, useful for the development of a predictive QSAR model. The application of 3D-QSAR methods has been used to evaluate the inhibition of CYP3A4 by dietary compounds such as anthocyanins [69] and flavonoids [70].

### 2.3. Protein–Ligand Interactions

An X-ray structure or model of the interaction of a selected compound with the CYP enzyme of interest can often enhance aspects of our general understanding of CYP binding that cannot be illuminated by ligand-based methods. Kiani et al. (2019) [71] used docking and MD simulation protocols to explore the role of binding-pocket residues in the CYP3A4-inhibitor binding mechanism. Their work demonstrated the importance of the binding-site residues Arg106 and Arg372 in the stabilization of the CYP3A4-inhibitor complex.

Docking methods aim to predict the orientation of a ligand when it is bound to a protein. To achieve this, measures of shape, van der Waals and electrostatic interactions, and the formation of hydrogen bonds are calculated. Therefore, docking is often used to identify the following key features of the CYP–inhibitor interaction: ligand orientation, the involvement of specific amino acid residues, and proximity to the heme; moreover, it is used to rationalize and support experimental findings regarding DDI, FDI and HDI [72,73,74]. For example, docking studies showed that ginger (Zingiber officinale) components interact with an array of amino acids in the active sites of CYP1A2, CYP2C9, CYP2C19, CYP2D6 and CYP3A4, mainly through the formation of hydrogen bonds and, to a lesser extent, via π–π stacking [75]. 

The calculated energies of these interactions are summarized as a docking score, which represents the potentiality of binding. These scoring methods were developed as a trade-off between screening speed and accuracy. Hence, docking scoring is efficient and performs well for rough preliminary filtering of candidate ligands and for the comparison of congeneric series of molecules, but is not particularly accurate. Induced-fit docking (IFD), an advanced docking approach, takes into account the flexibility of both the protein and the ligand, improving the fit of the ligand at the binding site and the energetic terms calculated for the complex [76]. Since this method is more expensive in terms of computational power and time, it is usually implemented for pose optimization in small-scale projects rather than high-throughput screening. For example, IFD was used to evaluate the potential of four novel compounds to inhibit five CYP isozymes [77]. IFD was also used to explain the different efficiency in the oxidation of aristolochic acid I by various CYP isozymes [78]. 

At the other end of the spectrum, highly accurate methods to calculate the free energy of ligand binding have been developed. These methods are based on extensive Monte Carlo sampling or MD simulations of the complex and the free ligand in solution. Such computationally intensive calculations often make these methods inapplicable in FDI research [79]. 

Between these extremes are the end-point methods, of which MM/PBSA (molecular mechanics Poisson–Boltzmann surface area) and MM/GBSA (molecular mechanics generalized Born surface area) are the most widely used. As the name indicates, end-point methods are based on samplings of the final states of a system, that is, the complex and the free receptor and ligand. Therefore, they are much less computationally expensive than the pathway methods and more accurate than most docking–scoring functions. The MM/GBSA method was implemented by Kehinde et al. (2020) [80] to calculate the binding affinity of epigallocatechin gallate, kaempferol-7-glucoside, luteolin, and ellagic acid to CYP3A4, and later, specific ligand–protein interactions were assessed using docking. In previous work conducted by our group, MM-GBSA calculations differentiated between high- and low-potency CYP3A4 inhibitors and supported in vitro findings of CYP3A4 inhibition by furanocoumarins [73].

## 3. Ligand-Based Methods

Ligand-based methods aim to screen for new inhibitors and have been continuously adapted to identify the molecular features of ligands that contribute to their inhibition capacity. As implied by the name, these methods are independent of protein structure and require accumulated data regarding the in vitro and in vivo inhibition activities of the compounds. However, ligand-based 3D methods, such as pharmacophore modeling, also require data about the ligands’ bound conformation and position at the binding site, so they rely on protein structure, at least partially, as demonstrated by Hochleitner et al. (2017) [81]. Those researchers filtered compounds from databases of natural products that share similar pharmacophore features with established inhibitors [81]. 

Recently, the rapid expansion in high-throughput in vitro assays has increased the availability of evidence-based data. Consequently, and with the widespread use of deep learning and other machine-learning algorithms, the emphasis of in silico studies related to CYP inhibition has shifted from explanatory modeling toward ligand-based QSAR. Ligand-based QSAR relies on the generalization of the chemical-structure information of compounds correlated to experimental data concerning protein binding and inhibition. In general, machine-learning methods can be classified into two groups: regression, based on the use of numerical data (i.e., the IC_50_ of a CYP inhibitor) and classification, based on the use of discrete classes of inhibition potency. The vast majority of models published in the last decade fall into the second category. Despite being a powerful approach for predicting new inhibitors, ligand-based calculations and predictions are difficult to interpret, especially in models that use fingerprints rather than 2D or 3D descriptors. 

A large dataset of more than 24,700 unique compounds, extracted from PubChem, was used to develop prediction classifiers for five major CYP isoforms—CYP1A2, CYP2C9, CYP2C19, CYP2D6 and CYP3A4—by Cheng et al. [82]. They fused various independent deep-learning and other machine-learning classifiers, including a support vector machine, a C4.5 decision tree, k-nearest neighbor and naive Bayes, in a back-propagation artificial neural network. The area under the receiver operating characteristic curve (AUC) for the validation sets of the developed models was in the range of 0.764 to 0.815 for CYP1A2, 0.837 to 0.861 for CYP2C9, 0.793 to 0.842 for CYP2C19, 0.839 to 0.886 for CYP2D6 and 0.754 to 0.790 for CYP3A4. However, when the classifiers were validated against a subset of molecules outside the applicability range of the training set, they did not perform as well. The performance of the combined classifiers fused with a back-propagation artificial neural network was superior to that of three classic fusion techniques, as well as that of any individual classifier.

The same PubChem dataset was also used by Sun et al. (2011) [83] to develop support-vector classification models and by Lapins et al. (2013) [84] to develop the Bioclipse Decision-Support open-source system, with somewhat improved AUC results. Similar results were later obtained from a multiple-category classification model, which was built for the five CYP isoforms using a Laplacian-modified naive Bayesian method [85].

With Cheng and colleagues’ (2011) [82] high starting point, the models developed later present various methods and algorithms, but only minor improvements in accuracy and prediction power. Given the similarity between different CYP isoforms and the similarity between inhibitors, Li et al. (2018) [86] showed the minor advantage of a multitask model that can predict inhibitors for five isozymes at the same time, as compared to five independent models. The predictive power of the various models is determined not only by the choice of algorithm, but also by the parameters of implementation, such as data pre-processing, the choice of descriptors and the size of the training dataset [87]. Often, in vitro data and other measurement data available in public databases have been obtained from different studies involving different protocols or experimental conditions, such as protein concentration, incubation time, time points, buffer composition, etc. In such cases, intensive data curation improves model performance [88].

Another important factor that affects the performance of classification models is the balance or imbalance of the classes (i.e., the variation in the number of samples in each class). To deal with class imbalance, several approaches have been suggested for over-sampling the minority class or under-sampling the majority class [89,90,91]. One of the approaches that has been proven to be effective for both small and large datasets is under-sampling by clustering the majority class into a number of clusters that is the same as the number of samples in the minority class [92].

The above-described models can be used to virtually screen natural products, including phytochemicals, compounds of marine and microbial origin and dietary compounds, to identify new inhibitors. However, to the best of our knowledge, few-to-no results of such screening have been published, demonstrating a large bias of focus toward DDI rather than FDI. 

A deep-learning classifier was built by the authors to predict CYP3A4 inhibitors. The classifier was built using Deepchem, a deep-learning algorithm, based on the convolutional neural-network method. The virtual screening of ~60,000 dietary compounds from the FooDB by the classifier revealed the presence of 115 dietary CYP3A4 inhibitors [93].

## 4. Databases

The rapid progress in machine-learning methods and their implementation for chemo-informatics goes hand-in-hand with the accumulation of data collected through experiments. Most of these data have been collected, annotated and published for public use. In the following section, we will review the databases relevant for FDI analysis.

### 4.1. In Vitro Inhibition Data

The PubChem BioAssay database contains bioactivity data regarding compounds that have been tested in vitro against various CYPs. The major bioactivity datasets that have been compiled from high-throughput screening assays are summarized in Table 1. Using these bioactivity data, several groups have developed computational models for predicting CYP inhibition by small molecules, as reviewed above [82,83,85,86]. AID1851 is a dataset that includes in vitro data regarding inhibitors of the five major CYP isoforms; together, these account for over 90% of drug metabolism, namely, CYP1A2, CYP2C9, CYP2C19, CYP2D6 and CYP3A4. AID1851 is the most commonly used dataset of this kind. Other CYP inhibition-related bioassay datasets are limited to single CYP isoforms. The majority of the substances are shared between AID1851 and the corresponding single-isoform dataset (i.e., AID410, AID 883, AID899, AID891 and AID884 for CYP1A2, CYP2C9, CYP2C19, CYP2D6 and CYP3A4, respectively; Table 2). The high correlation (0.937–0.999) between duplicates in related datasets confirms the reliability of the data.

In AID885, the same set of compounds as that used in AID884 was assayed for CYP3A4-activator activity. CYP activators are much less prevalent and studied, although their effects on CYP activity are no less important. Three new datasets (AID1645840-2) were released in March 2021 with 5242 substances that were not assayed for inhibition activity in older HTS bioassays. These new data offer a foundation for the development of new models with possibly extended applicability domains.

### 4.2. Dietary Compounds

The increasing availability of large datasets of dietary compounds is both timely and vital for the virtual screening for FDIs. Such interactions mediated by CYPs are usually related to secondary metabolites, which are generally smaller than 1500 Da and exhibit great chemical variability. These metabolites are consumed as food, beverages, spices, dietary supplements and traditional herbal remedies.

Despite the accumulating data regarding natural products, there is no global comprehensive database to which their structures and annotations can be submitted and queried. Currently, there are numerous partial databases, both open and commercial, with different scopes and structures and with different querying methods. Recently, Sorokina and Steinbeck (2020) [97] published a large comprehensive review of 123 natural-product databases and released the COlleCtion of Open Natural prodUcTs (COCONUT), an online aggregated dataset of natural products assembled from these open resources [98]. 

A database suitable for virtual screening must provide retrievable structural information regarding the molecules and, preferably, some metadata, as well (i.e., chemical classification, biological source, average concentration, etc.). The largest free databases that meet these criteria are the FooDB (70,926 dietary compounds, as of 15 May 2022), Collective Molecular Activities of Useful Plants (CMAUP, 47,645 natural-product molecules, as of 15 May 2022) and the Natural Product Activity and Species Source Database (NPASS, 35,032 natural-product molecules, as of 15 May 2022). CHEMnetBASE is a commercial database that includes ~85,000 natural products from food (Dictionary of Food Compounds) and ~65,000 marine natural products (Dictionary of Marine Natural Products). AntiMarin (>60,000 compounds) and MarineLit (>29,000 compounds) are two other large commercial databases of marine natural products. Phenol-Explorer is a comprehensive database dedicated to polyphenols found in foods. 

Several databases collect natural products from plants endemic to specific geographic areas. The African Natural Products Database (ANPDB) contains ~6000 natural products from northern and eastern Africa. TIPdb includes ~8500 natural products from plant species endemic to Taiwan, and VIETHERB is a collection of ~10,000 metabolites from herbs that are used in traditional Vietnamese medicine. The TCM database is currently the largest and most comprehensive free small-molecular database of substances that are important in traditional Chinese medicine that can be used for virtual screening. The geographic distribution of natural products is especially interesting for CYP-related FDI, as CYP genetic variation has been shown to be related to ethnicity [53,99,100].

## 5. Concluding Remarks

This review aims to establish the need for and demonstrate the number and diversity of available computational methods for the study of the effects of food, herbs and traditional medicines on medical regimes, by focusing on the modification of CYP-dependent drug metabolism. Structure- and ligand-based approaches have been widely validated for exploring a wide range of theoretical and practical questions. To date, the study of FDI and HDI has involved mostly the structure-based methods. Docking, as a user-friendly, accessible and easily interpretable approach, leads this trend. However, classic docking ignores the flexibility of the protein and the ligand, their adjustment to each other during binding and the influence of their aquatic environment. Other more advanced methods, such as MD and flexible docking (i.e., IFD) have been developed to fill this gap. 

Ligand-based machine-learning approaches are, by far, the most commonly used methods for virtual screening. However, while virtual screening is widely used to predict pharmaceutical CYP inhibitors and DDI, there has been very little screening for dietary compounds and natural products that may be responsible for FDI. This can be clearly seen by conducting a simple literature search. Querying the SciFinder^n^ (CAS, Columbus, OH, USA) library of publications with “DDI” and “CYP” yields 1638 manuscripts since 1995, while the query “FDI” and “CYP” yields only 133, a tenth of that (as of 15 May 2022). Since the number of commercially available natural products is orders of magnitude smaller than the diversity that exists in nature, virtual screening presents great potential for use with natural products. 

The predictive performance of the QSAR models reviewed here is strong and probably approaches the limit of virtual tools, based on the high quality of the available in vitro high-throughput screening data. Similar strong classification performance has been reported for new algorithms that are being developed and disseminated. As a further step, it would be useful to include an evaluation of the models’ applicability domains, that is, the scope of their limitations concerning their respective physicochemical domains [101]. With attention paid to applicability domains, ligand-based, machine-learning QSAR models can be carefully used to screen compounds from existing databases of natural products. 

This work highlights the need for a comprehensive open database of natural products that will be commonly recognized and allow the easy submission of newly found molecules and easy access to the stored data. The recently established COCONUT database aims to meet this challenge. The rapidly accumulating biological data regarding CYP activity and inhibition, CYP structure and natural products clearly gives rise to interesting research questions and hypotheses, as well as practical solutions. It is hoped that this review will be useful in focusing readers’ attention on those methods that can be readily applied to the study of FDI.

## Figures and Tables

**Figure 1 ijms-23-08498-f001:**
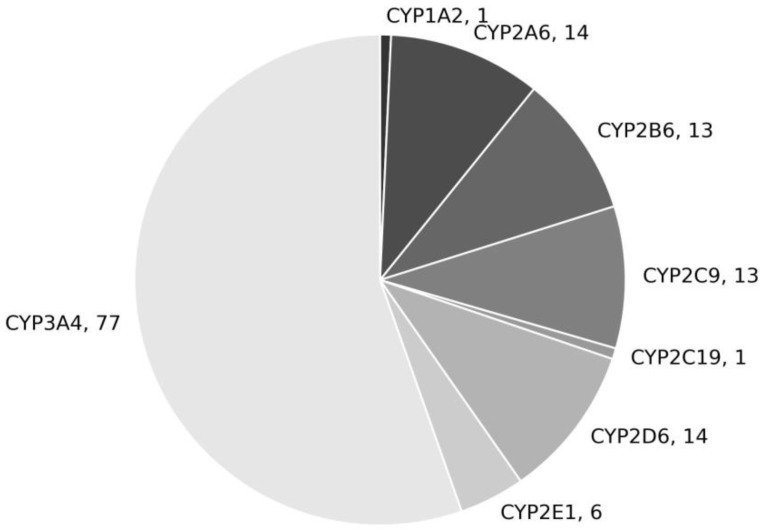
The distribution of structures of human CYP isoforms in the Protein Data Bank, as of May 2022.

**Table 1 ijms-23-08498-t001:** Major datasets of CYP isoforms in the PubChem BioAssays database.

CYP	Bioassay	Compounds	Substances	References
CYP1A2	410	8354	9198	[82,86,94]
CYP2C9	883	9385	10,320	[82,86]
	1,645,842	5094	5242	-
CYP2C19	899	9385	10,320	[82,86]
CYP2D6	891	9385	10,320	[81,82,86]
	1,645,840	5094	5242	-
CYP3A4	884	13,076	14,155	[82,86]
	885	13,076	14,155	[82]
	1,645,841	5094	5242	
CYP1A2, CYP2D6, CYP2C9, CYP2C19, CYP3A4	1851	16,560	17,143	[82,83,85,86,87,94,95,96]

**Table 2 ijms-23-08498-t002:** Similarity of AID datasets to AID1581.

	CYP1A2	CYP2C9	CYP2C19	CYP2D6	CYP3A4
Substances	9198	10,320	10,320	10,320	14,155
Shared substances with AID1581	7942 (86%)	9124 (88%)	9124 (88%)	9124 (88%)	9124 (64%)
Pearson’s correlation coefficient of activity scores	0.937	0.996	0.995	0.999	0.994

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
