# Peer review of "Computer-Aided (In Silico) Modeling of Cytochrome P450-Mediated Food–Drug Interactions (FDI)"

_ijms, 2022, doi:10.3390/ijms23158498_

Round 1

Reviewer 1 Report

This is a very fascinating paper that reviews recent developments in the P450 field. They particularly mention how food ingredients clash with drug molecules during P450 activation and effect drug effectiveness. I was not aware of this. Overall the review reads well and is structured well and fits the remit of this journal.

Some minor points:

1. I would add a short description of the general catalytic cycle of the P450s and what type of reactions they catalyze.

2. The introduction mentions food ingredients clashing the drug molecules in P450 activation but no results are shown, please add.

3. Abstract & Intro line 1: “cytochrome peroxidase P450”; this expression is wrong, the P450s use O2 rather than H2O2 and therefore are “cytochrome P450 mono-oxygenases” 

Author Response

Reviewer 1

We thank the reviewer and are encouraged by his evaluation of our manuscript.

Modifications:

  1. Rev: “I would add a short description of the general catalytic cycle of the P450s and what type of reactions they catalyze”.

Au: we accept the reviewer’s comment and added the following paragraph in the introduction: “The catalytic process of CYPs involves the insertion of one atom of oxygen into an organic substrate while another oxygen atom is reduced to water.”

  1. Rev: “The introduction mentions food ingredients clashing the drug molecules in P450 activation but no results are shown, please add”.

Au: we added specific compounds to the foods provided as examples in the 3rd paragraph. However, while the inhibiting compounds in some foods were identified, for others, the specific active compound is still unknown. We also added this explanation in the manuscript.

  1. Rev: “Abstract & Intro line 1: “cytochrome peroxidase P450”; this expression is wrong, the P450s use O2 rather than H2O2 and therefore are “cytochrome P450 mono-oxygenases”.

Au: we agree with the reviewer and therefore changed the sentence in the abstract as follows: “…activity of Cytochrome P450 (CYP) enzymes…” and in the introduction as follows: “Cytochrome (CYP) 450 enzymes are heme mono-oxygenases…”

We would like to thank again the reviewer for providing us with the opportunity to improve the clarity of our manuscript. 

Sincerely,

Zohar Kerem, Corresponding author

Reviewer 2 Report

The authors reviewed computational tools and approaches especially by dividing it into two categories, structure-based and ligand-based methods, to predict CYP-mediated FDI. This manuscript is clear and reasonable for current status of computer-aided approaches or limitation. Only some minor revisions are needed as mentioned below.

1.       Introduction 4th paragraph : The isoform CYP3A4 is not expressed in rodents because of species differences, for example, Cyp3a11, Cyp3a13, … in mice, Cyp3a1, Cyp3a2, … in rats. In addition, the sentence “with only scant expression in the intestine” is not correct. Cyp3a11 is significantly expressed in the intestine of mice, and the CYP3A expression pattern is largely different between mice and rats. The first sentence should be revised.

 2.       Academically incorrect terms should be avoided.

Introduction line 1 : Cytochrome peroxidase (CYP) 450 enzymes cytochrome P450 (CYP)

I would suggest not to omit “CYP” when describing CYP isoforms.

Ex) CYP1A2, 2C19, 3A4 CYP1A2, CYP2C19, CYP3A4

Author Response

We thank the reviewer for a thorough review and helpful comments. We are happy that the reviewer found our work interesting to the JAFC audience.

Modifications:

  1. Rev: “Introduction 4th paragraph: The isoform CYP3A4 is not expressed in rodents because of species differences, for example, Cyp3a11, Cyp3a13, … in mice, Cyp3a1, Cyp3a2, … in rats. In addition, the sentence “with only scant expression in the intestine” is not correct. Cyp3a11 is significantly expressed in the intestine of mice, and the CYP3A expression pattern is largely different between mice and rats. The first sentence should be revised.”.

Au: we accept the reviewer’s comment and omitted the second part of the sentence.

  1. Rev: “Academically incorrect terms should be avoided.

Introduction line 1: Cytochrome peroxidase (CYP) 450 enzymes → cytochrome P450 (CYP)

I would suggest not to omit “CYP” when describing CYP isoforms.

Ex) CYP1A2, 2C19, 3A4 → CYP1A2, CYP2C19, CYP3A.”

Au: we accept the reviewer’s comment and revised the mentioned terms as follows:

Line 1: “Cytochrome (CYP) 450 enzymes are heme mono-oxygenases…”,

“CYP” was added to the description of CYP isoforms

We would like to thank again the reviewers and the editor for providing us with the opportunity to improve the clarity of our manuscript.

Sincerely,

Zohar Kerem, Corresponding author